# Blood and Sputum Eosinophils of COPD Patients Are Differently Polarized than in Asthma

**DOI:** 10.3390/cells12121631

**Published:** 2023-06-15

**Authors:** Katarzyna Mycroft, Magdalena Paplińska-Goryca, Małgorzata Proboszcz, Patrycja Nejman-Gryz, Rafał Krenke, Katarzyna Górska

**Affiliations:** Department of Internal Medicine, Pulmonary Diseases and Allergy, Medical University of Warsaw, 02-097 Warsaw, Poland; katarzyna.mycroft@wum.edu.pl (K.M.); malgorzata.proboszcz@wum.edu.pl (M.P.); patrycja.nejman-gryz@wum.edu.pl (P.N.-G.); rafal.krenke@wum.edu.pl (R.K.); katarzyna.gorska@wum.edu.pl (K.G.)

**Keywords:** asthma, COPD, eosinophils, inflammation, sputum

## Abstract

Different eosinophil subpopulations have been identified in asthma and other eosinophilic disorders. However, there is a paucity of data on eosinophil subpopulations in patients with chronic obstructive pulmonary disease (COPD). The aim of this study was to compare eosinophil phenotypes in blood and induced sputum in patients with COPD, asthma and controls. Stable patients with mild-to-moderate COPD (*n* = 15) and asthma (*n* = 14) with documented blood eosinophilia ≥100 cells/µL in the year prior to the study and the control group (*n* = 11) were included to the study. The blood and sputum eosinophil phenotypes were analyzed by flow cytometry. IL-5, IL-13, CCL5 and eotaxin-3 levels were measured in the induced sputum. The marker expression on blood eosinophils was similar among control, asthma and COPD groups. The expressions of CD125, CD193, CD14 and CD62L were higher on blood than on sputum eosinophils in all three groups. We found increased levels of CD193+ and CD66b+ sputum eosinophils from COPD patients, and an elevated level of CD11b+ sputum eosinophils in asthma compared to COPD patients. The results of our study suggest that the profile of marker expression on COPD sputum eosinophils differed from other groups, suggesting a distinct phenotype of eosinophils of COPD patients than in asthma or healthy subjects.

## 1. Introduction

Eosinophils are crucial in the development of allergies and other inflammatory conditions, and in fighting parasitic infections. Eosinophils originate from the bone marrow and migrate through the bloodstream to different organs and tissues, e.g., lymph nodes, gastrointestinal tract, spleen and lung [1,2]. Eosinophil maturation, migration and recruitment are regulated by cytokines and chemokines produced locally at inflammation sites: eotaxin-1, -2, -3; interleukin (IL)-4, IL-5, IL-33; granulocyte-macrophage colony-stimulating factor (GM-CSF); regulated on activation, normal T-cell expressed and secreted (RANTES); and surface adhesion proteins (e.g., complement receptors CR3–CD11b/CD18 or L-selectin–CD62L) expressed on eosinophils [3,4]. Under certain conditions, e.g., in response to allergens or parasites, the production of mediators attracting eosinophils is enhanced by T helper type 2 (Th2 cells), leading to eosinophil infiltration of the tissue [3,5]. In some cases, the eosinophilic inflammation may become uncontrollable through the activity of factors promoting survival or inhibiting apoptosis (IL-3, -5, GM-CSF, alarmins) [6,7,8], and increased eosinophil recruitment. Eosinophilic inflammation is the cornerstone of the pathogenesis of asthma, chronic rhinosinusitis and eosinophilic esophagitis [3]. The prolonged eosinophil accumulation in the airways leads to uncontrolled eosinophil degranulation and the release of cytotoxic proteins (major basic protein (MBP), eosinophil cationic protein (ECP), eosinophil peroxidase (EPX), eosinophil-derived neurotoxin (EDN)) from granules and, consequently, airway epithelium damage and impairment.

Elevated eosinophil levels are found in around one-third of COPD patients [9]. The exact role of eosinophils in COPD remains unknown but it has been shown that increased eosinophil levels are associated with better treatment response to inhaled corticosteroids [10,11,12]. However, inhaled corticosteroid therapy in eosinophilic COPD patients is not as effective as in asthma [13].

Further discrepancies between COPD and asthma have been observed when evaluating treatment responses to therapies directed against IL-5, which have been shown to be effective in severe eosinophilic asthma [14]. However, the effectiveness of IL-5 biologics in eosinophilic COPD patients was found to be worse than in asthma [15]. On the other hand, the results of the recent BOREAS trial have shown that dupilumab, a monoclonal antibody blocking the shared receptor component for interleukin-4 and interleukin-13, is effective in both eosinophilic COPD and asthma patients [16]. We hypothesize that the molecular regulation of eosinophil function differs in these two obstructive lung diseases.

Some authors identified different eosinophil subpopulations in asthma and other pathological conditions [2]. Sputum eosinophils of asthma patients were CD62L^low^IL-3R^high^, while in non-asthmatic patient lungs, eosinophils were CD62L + IL-3R^low^ [17]. Differences were also observed between circulating and airway eosinophils, suggesting distinct activation states. CD11b, CD18 and CD29 (Integrin beta 1) were upregulated while CD193 (CC-chemokine receptor 3, CCR3), CD125 (Interleukin-5 receptor alpha) and CD62L were downregulated in local airway eosinophils compared to systemic blood eosinophils [18]. However, there is a paucity of data on eosinophil subpopulations in COPD patients. We hypothesize that eosinophils in COPD and asthma differ in their surface marker expression. Therefore, the aim of this study was to assess eosinophil phenotype variants in the blood and in the induced sputum of COPD, asthma and control subjects.

## 2. Materials and Methods

### 2.1. General Study Design and Participants

Stable patients with mild-to-moderate COPD or asthma, presenting to the outpatient clinic at the Department of Internal Medicine, Pulmonary Diseases and Allergy of the Medical University of Warsaw from October 2020 to February 2022, were enrolled in the study. The study was approved by the ethics committee of the Medical University of Warsaw (KB/62/A/2020). All participants provided informed consent. This study was registered on ClinicalTrials.gov, number NCT05398133.

Inclusion criteria for the COPD group were as follows: (1) age 40 years and over; (2) diagnosis of COPD based on past medical history, history of smoking ≥ 10 packyears, typical signs and symptoms [19] and irreversible airway obstruction found in spirometry (z-score of the post-bronchodilator FEV1/FVC below −1.645 [20]); (3) blood eosinophilia ≥ 100 cells/µL in the year prior to the study.

Inclusion criteria for asthma were as follows: (1) age 18 years and over; (2) diagnosis of asthma based on past medical history, typical signs and symptoms and demonstration of variable expiratory airflow limitation [21]; (3) a negative smoking history; (4) blood eosinophilia ≥ 100 cells/µL in the year prior to the study.

Patients were excluded if there was a documented current or previous history of asthma (for COPD patients) or COPD diagnosis (for asthma patients); history of any other chronic lung disease, autoimmune and hematological diseases, malignancies, severe cardiovascular diseases; COPD or asthma disease exacerbation requiring treatment with systemic corticosteroid and/or antibiotics in the past 3 months; biological treatment (e.g., mepolizumab); history of using immunosuppressives; or current helminth infection.

Control subjects were matched by age and gender to the study groups and were recruited among volunteers with normal spirometry who had no lung condition, hematological or malignant disease, and had no respiratory tract infection in the past 3 months.

Clinical and demographic data (including exacerbation history, comorbidities, medication) were collected during the visit. COPD patients performed the COPD Assessment Test (CAT) and modified Medical Research Council (mMRC) dyspnea scoring, while asthma patients completed the Asthma Control Test (ACT). Spirometry was performed according to the ATS/ERS guidelines [22]. Atopy was diagnosed when either at least one skin prick test for inhalant allergens was positive (a mean wheal diameter ≥ 3 mm) or the specific IgE level for inhalant allergens was over 0.35 kU/L.

### 2.2. Blood and Sputum Sampling and Processing

Venous blood was taken for measurement of a full blood count, serum C reactive protein (CRP), total immunoglobulin E (IgE), inhalant allergen panel, cytokine measurements and flow cytometry. Peripheral blood mononuclear cells (PBMCs) and granulocytes were isolated from freshly drawn venous blood (9 mL collected in EDTA tube) by Lymphoprep (StemCell, Vancouver, BC, Canada) centrifugation according to manufacturer protocol.

Sputum induction was preceded by inhalation of 400 μg of salbutamol. Then, inhalations of sterile hypertonic saline (NaCl) were applied at increasing concentrations (3, 4, and 5% solutions) via an ultrasonic nebulizer (ULTRA-NEBTM2000, DeVilbiss, Somerset, PA, USA) following ERS recommendations [23]. The induced sputum samples were processed as described previously [24].

### 2.3. Assessment of Surface Receptor Expression by Flow Cytometry

Cells were stained with antibodies against the surface binding molecules: CD45 (mouse anti-human APC-H7, cat. no. 641408), CD16 (mouse anti-human BV510, cat. no. 563830), CD125 (mouse anti-human BV421, cat. no. 743927), CD193 (mouse anti-human PE, cat. no. 558165), CD62L (mouse anti-human BV605, cat. no. 562719), CD66B (mouse anti-human PerCP-Cy5-5, cat. no. 562254), CD11B (rat anti-human APC, cat. no. 564985) and CD14 (mouse anti-human Alexa Fluor 488, cat. no. 557700) (BD Biosciences, San Jose, CA, USA) in BD Horizon Brilliant Stain Buffer (BD Biosciences, San Jose, CA, USA), and incubated for 20 min in the dark at room temperature. Cells were analyzed on a FACS Celesta flow cytometer (BD Biosciences, San Jose, CA, USA) equipped with blue (488 nm), violet (405 nm) and red (640 nm) lasers. Unstained cells and compensation beads (BD Biosciences, San Jose, CA, USA) were used to set voltages and create single-stain negative and positive controls. Compensation was set to account for spectral overlap between the seven fluorescent channels used in the study. The cut-off gating for negative populations was determined using fluorescence minus one (FMO) (a sample that has been stained with all fluorophores except one) controls (Figure 1A). Samples were examined by side scatter area (SSC-A) vs. forward scatter area (FSC-A), then using forward scatter height (FSC-H) vs. FSC-A to select single cells, eliminating debris and clumped cells from the analysis. At least 50,000 cells in the target gate were collected. The eosinophils were identified as SSChi, CD45+ and CD16− cells. In the first step, each marker (CD125, CD193 CD14, CD11b, CD62L, CD66b) was analyzed separately in each group. Next, the eosinophil subpopulations were characterized by the expression of the following pairs of markers: (1) CD125 and CD193, (2) CD14 and CD11b, (3) CD62L and CD66b. BD FACS Diva 8.0.1.1 software was used for data analysis. The scheme of the gating strategy for blood and sputum is presented in Figure 1B,C.

### 2.4. Protein Analysis in Induced Sputum

Sputum cytokine and eotaxin levels were measured by using ELISA kits (R&D Systems, Minneapolis, MN, USA) according to the manufacturer’s instructions. The minimal detectable concentrations were as follows: IL-5 (0.29 pg/mL), IL-13 (13.2 pg/mL), RANTES (2.0 pg/mL), eotaxin-3 (0.215 pg/mL).

### 2.5. Statistical Analysis

Statistical analyses were performed using Statistica 13.3 software (StatSoft Inc., Tulsa, OK, USA) and GraphPad Prism software (version 9.3.1) (GraphPad Software, Inc., San Diego, CA, USA). Data are presented as median and interquartile range (IQR) or number and percentage. Differences between continuous variables in 3 groups were tested using the Kruskal–Wallis test and between 2 groups using the nonparametric Mann–Whitney U test. For the comparison of surface marker expression between blood and sputum eosinophils, the Wilcoxon matched pairs test was used. Fisher’s exact test was used to test the differences between nominal variables. Correlations between variables were analyzed with Spearman’s rank test. A *p*-value of less than 0.05 was taken as the threshold of statistical significance.

## 3. Results

### 3.1. Patient Characteristics

In total, 11 controls, 14 asthmatics and 15 COPD patients were recruited to the study; blood and induced sputum were collected from all study participants (Figure 2). Patients’ characteristics are presented in Table 1. COPD patients were significantly older than asthma patients, but there were no differences in age between COPD patients and controls and between asthma and control subjects. Among COPD patients, 5 were ex-smokers and 10 were current smokers. In the control group, five patients were smokers. Both asthma and COPD patients were mildly to moderately symptomatic; the median ACT and CAT scores were 20 (14–23.5) and 14 (8–21) for asthma and COPD patients, respectively. At the time of the study, two patients with COPD and five patients with asthma received low-dose ICS.

### 3.2. Expression of CD125, CD193, CD62L, CD66b, CD14, CD11b on Sputum and Blood Eosinophils

Data on the total percentage of sputum and blood eosinophils with surface CD125, CD193, CD62L, CD66b, CD14 and CD11b expression in COPD, asthma and control subjects are presented in Figure 3. The profiles of blood and sputum eosinophils differed significantly. We found that blood eosinophils were characterized by increased expression of CD125+, CD193+, CD62L and CD14+ compared to sputum cells in all evaluated groups, suggesting the changed phenotype of eosinophils after recruitment into the airway. Interestingly, the expression of CD11b was increased on sputum 99.4 (98.8–100.0)% compared to blood 96.9 (82.6–98.8)% eosinophils (*p* = 0.002), but only in the asthma group. The expression of CD66b remained the same in systemic and airway eosinophils, implying the important role of CD66b as a unified eosinophil marker.

We found that the profile of marker expression on COPD sputum eosinophils differed from other groups. We showed an increased percentage of CD193+ sputum eosinophils from COPD patients compared to controls (48.1 [29.8–69.3]% vs. 18.7 [11.6–32.8]%, respectively, *p* = 0.028), a higher proportion of CD66b+ sputum eosinophils in COPD subjects compared to controls (67.9 [59.9–75.7]% vs. 39.1 [25.6–58.4]% and in asthma compared to controls (*p* = 0.024), respectively, *p* = 0.036), and a decreased level of CD11b+ sputum eosinophils in COPD compared to asthma (97.9 [96.6–99.1]% vs. 99.4 [98.9–100]%, respectively, *p* = 0.009).

**Eosinophil phenotype analysis based on the expression of CD193 and CD125.** Blood and sputum eosinophils in all three groups were characterized by the predominance of the CD193+CD125+ profile. Interestingly, we found a decreased level of CD193−CD125+ sputum eosinophils in COPD compared to controls (Figure 4).

### 3.3. Cytokine and Eotaxin Concentrations in Sputum Samples

Protein sputum levels are presented in Table 2. There were no differences in concentrations of IL-5 and eotaxin-3 among the groups.

### 3.4. Correlations between Th2 Inflammation Biomarkers and Eosinophil Phenotype

There was no correlation between blood and sputum eosinophils and IL-5 and eotaxin-3 concentrations in any of the examined groups.

We found no correlations between sputum IL-5 concentration and CD125 expression on sputum or blood eosinophils and between sputum eotaxin-3 concentration and CD193 expression on sputum or blood eosinophils. Moreover, there was no correlation between either IL-5 or eotaxin-3 and CD125+CD193+ eosinophil subpopulation percentage in any of the examined groups.

Total IgE level correlated with sputum CD125−CD193+ eosinophil percentage in the whole group (r = 0.44, *p* = 0.041). There was a strong correlation between IgE level and sputum CD125 expression in the asthma group (r = 0.79, *p* = 0.036) which was not found in COPD or controls. IgE level did not correlate with IL-5 or eotaxin-3 concentrations or the expression of CD193 on sputum and blood eosinophils in asthma, COPD or control subjects.

## 4. Discussion

This study is, to our knowledge, the first that analyzed eosinophil subsets in the blood and sputum of COPD patients and compared them to asthma and control subjects. We found that the expressions of CD125, CD193, CD14 and CD62L were higher on blood than sputum eosinophils. We discovered a higher number of sputum CD193+ eosinophils and CD66b+ eosinophils in the COPD group compared to controls and CD11b+ eosinophils compared to the asthma group. The results of our study suggest different profiles of polarization of tissue and systemic eosinophils and a distinct phenotype of eosinophils in COPD patients compared to asthma or healthy subjects.

The eosinophil recruitment from systemic circulation to the local site of inflammation requires a series of biochemical and immunological changes in cells manifested by the expression of different surface receptors. The results of our study showed different profiles of surface markers between the blood and sputum eosinophils—we found an increased number of CD125+, CD193+, CD62L+ and CD14+ blood eosinophils compared to sputum eosinophils. Moreover, blood eosinophils’ marker expressions were similar across all three groups, suggesting that circulating eosinophils exhibit a low a priori activity level, irrespective of the pathological or normal conditions. Eosinophil activity in the airways depends on the stimulation of cytokines, mediators and antigens located in the lung. However, we found that different eosinophil subpopulations are present already in the blood, which is consistent with findings from another study [17].

The changed expression of surface markers on sputum eosinophils might reflect their increased activity under both normal and pathological inflammatory conditions [25]. According to our observations, bronchial eosinophils of COPD patients are characterized by a higher CD193 and CD66b expression compared to controls and CD11b expression compared to asthma. We believe that the local airway environment modulates the degree of activation of these cells. Upon reaching the airways, the eosinophils come into contact with locally produced mediators and become activated. The primed eosinophils in the airways participate in keeping the lung immune homeostasis and host defense [18]. These tasks are performed by a rapid eosinophil degranulation and the release of cytotoxic proteins and the formation of extracellular eosinophil traps which have the ability to control extracellular pathogens [26].

We observed that both CD193 and CD125 were highly expressed on blood eosinophils compared to sputum eosinophils in controls, COPD and asthma patients. The lower CD193 expression on sputum eosinophils initiated eotaxin-induced CD193 internalization and eosinophil activation in the airways [7]. Interestingly, sputum eosinophils in COPD patients had a higher expression of CD193 compared to controls. This finding is consistent with other studies that showed significantly greater proportions of airway cells expressing CCR3 in COPD and exacerbated chronic bronchitis patients compared with non-smoking controls [27,28]. The exact role of CD193 and CD125 in the pathogenesis of COPD remains unclear. It has been suggested that oxidative stress related to prolonged tobacco smoke exposure stimulates the Th2-like response in COPD pathobiology [29]. In a murine model of asthma, cigarette smoke induced allergic inflammation and was associated with delayed tolerance to inhalant antigens [30]. Cigarette smoke exposure alters the balance between Th1 and Th2 CD4+ T cells in the lung, which is mediated by lipocortin 1 [31]. The Th1 to Th2 shift promotes allergic sensitization whereas Th1 augmentation induces the development of emphysema [31]. In our study, the known eosinophil chemoattractants IL-5 and eotaxin-3 were found at similar levels in COPD, asthma and control subjects and did not correlate with CD125 and CD193 expression, respectively, suggesting other than the classical Th2-related pattern of eosinophil activation in the airways of COPD patients. Although no correlation between IL-5 and sputum eosinophils in asthma was found in other studies [32,33], sputum eotaxin-3 levels significantly correlated with sputum eosinophil counts in severe asthmatic patients [34]. This discrepancy between our results and other studies may be due to the difference in disease severity (no severe asthma in our study), as well as different cut-off points for the definition of eosinophilia (100 cells/µL vs. 300 cells/µL)

CD62L and CD11b are both adhesion molecules that participate in the migration of blood granulocytes to the place of local inflammation in the tissue [35]. CD62L has been shown to be downregulated on sputum eosinophils in asthma and healthy subjects [25,36]. Low CD62L expression was found on activated (“inflammatory”) eosinophils [17], which suggests that primed eosinophils lose their ability to migrate to other tissues. The CD62L is detached intravasally before granulocytes migrate from the blood to the airways [37]. Our results confirm that CD62L is shed from eosinophils migrating into the airways also in COPD. However, we did not find any differences in eosinophil CD62L surface expression between COPD, asthma and controls, indicating that altered expression of CD62L is not specific to airway inflammation in obstructive lung diseases. The downregulation of CD62L coincides with the upregulation of CD11b on activated granulocytes [25,35,37]. In our study, we only found increased expression in sputum eosinophils of asthma patients. Of note, the expression of CD11b on blood eosinophils was already high in all groups, which is a sign of eosinophil maturity [38] and suggests that circulating eosinophils are already primed.

The results of our study revealed that CD66b, a surface molecule involved in regulating the adhesion and activation of eosinophils [39], had a higher expression on sputum eosinophils of COPD and asthma patients compared to controls. The upregulation of CD66b in COPD and asthma might be caused by airway microbiome dysbiosis since a higher expression of CD66b on leukocytes has been linked to the homotypic clustering of cells during Gram-positive sepsis [40]. The cross-linking of CD66b by a natural ligand (galectin-3) causes tight cellular adhesion and clustering of CD11b and induces eosinophil effector functions [39]. Although we did not find any significant differences in the expression of CD66b between blood and sputum eosinophils, other authors found that CD66b was upregulated on sputum eosinophils in asthma [25]. In a recent study, CD66b^high^CD15^high^ eosinophil levels correlated with blood eosinophil number and IL-5 levels in sputum in allergic asthma with airway eosinophilia [41]. The increased CD66b expression in asthma probably reflects the inflammatory state of eosinophils; however, the role of CD66b in COPD eosinophils needs further evaluation.

The results of our study do not explain the differences in the response to the therapy with anti-IL5 biologicals between COPD and asthma patients, as the CD125 (IL-5 receptor) level was similar in both diseases. However, the differences in phenotypes between sputum eosinophils in the examined groups might indicate the future direction of personalized therapy in COPD and asthma, e.g., with CCR3 monoclonal antibody.

Our study has some limitations. The analysis of blood and sputum eosinophils in humans has many obstacles which have also been noted by other authors [42], e.g., small amounts of eosinophils compared to other cell types and eosinophil level variability. We included patients on the basis of blood eosinophilia, and all patients but one had low sputum eosinophils counts (<3%). Therefore, we could not characterize the phenotype of eosinophils in COPD patients with a marked sputum eosinophilia and compare it to those with low sputum eosinophil counts. This shortcoming could be limited by including patients based on sputum eosinophilia since blood and sputum eosinophil levels are not correlated [9]. It would also be interesting to study eosinophil phenotypes of severe COPD and asthma patients and to compare them with patients with mild-to-moderate disease. However, severe COPD and asthma patients are often treated with steroids which suppress inflammation. Moreover, we did not evaluate clinical features related to the Th2 pathway, such as bronchodilator responsiveness and fractional exhaled nitric oxide, which could have provided more insight into the study groups and the relationship between clinical and cellular data. Finally, we are aware that the study groups are relatively small. Despite the small study size, we found significant differences in eosinophil phenotypes between the study groups.

## 5. Conclusions

In conclusion, the results of our study suggest different profiles of polarization of airway and systemic eosinophils. The results of our study suggest distinct phenotypes of airway eosinophils in COPD compared to asthma and control patients.

## Figures and Tables

**Figure 1 cells-12-01631-f001:**
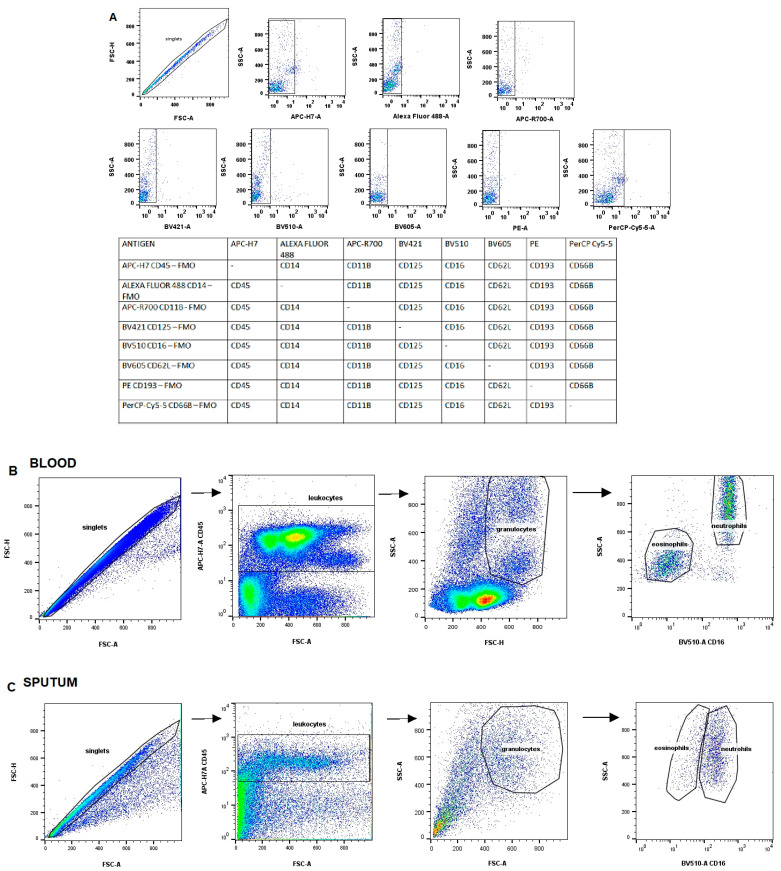
The gating strategy. (**A**) Fluorescent minus one (FMO) (**B**) blood, (**C**) sputum. Legend: Fluorescence minus one (FMO) controls for the experiment. FMO controls were prepared without adding a particular fluorophore-conjugated isotype control antibody, as shown in the table below. Gating strategy: Obtained events were gated in an FSC-A vs. FSC-H intensity dot plot to eliminate doublets. Leukocytes were gated on CD45+ vs. FSC. Granulocytes were selected as cells with high FSC and high SSC. The eosinophils were selected from granulocytes as CD16- cells. FSC: forward scatter, SSC: side scatter.

**Figure 2 cells-12-01631-f002:**
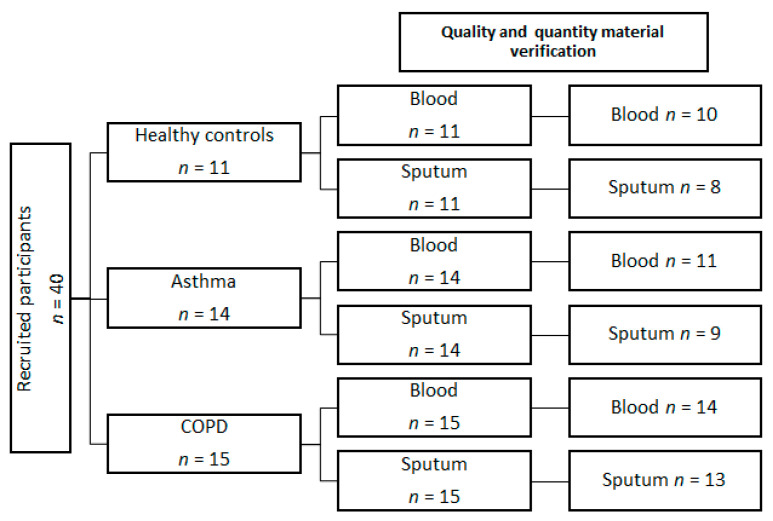
The flowchart of patients’ recruitment and study scheme. Forty patients were recruited; all patients had undergone blood sampling and sputum induction. Overall, 5 blood samples were not taken into FACS analysis because of low eosinophil levels and 10 sputum samples because of poor sputum quality (>20% of squamous cells, low number of cells) or low eosinophil levels. Blood samples were excluded if there were no eosinophils present in flow cytometry. The sputum was excluded if (1) the percentage of squamous cells was >20%, (2) there was a low number of cells isolated from sputum < 1 × 10^6^ or (3) there were no eosinophils in flow cytometry.

**Figure 3 cells-12-01631-f003:**
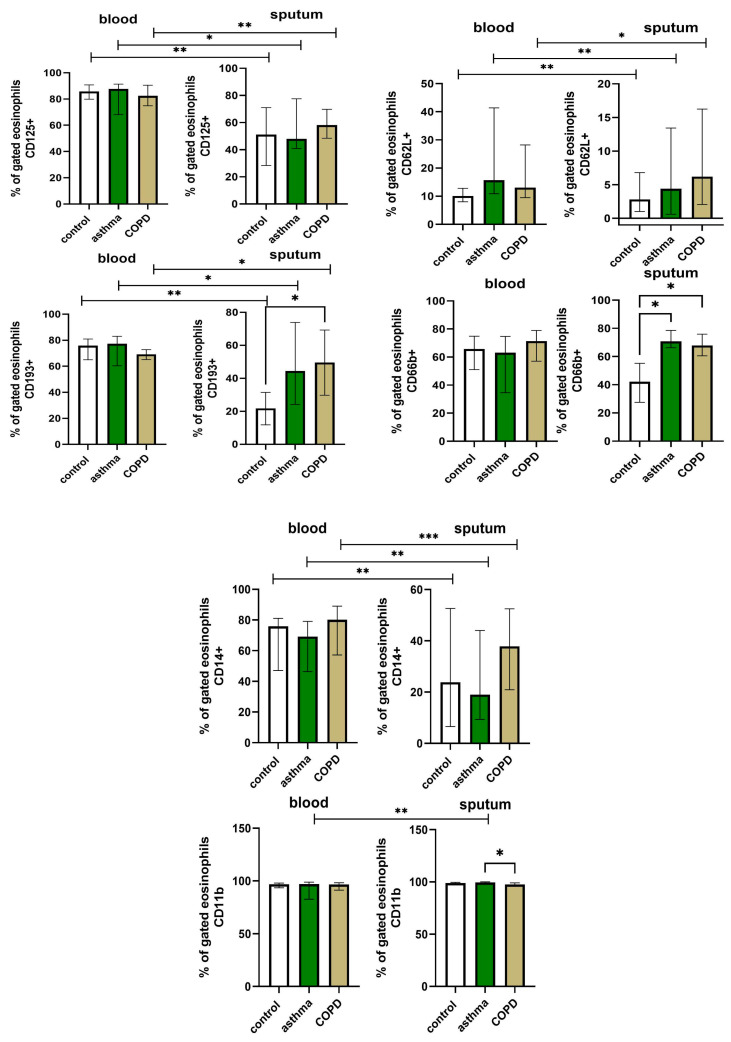
The comparison of CD125, CD193, CD62L, CD66b, CD14, CD11b expression on eosinophils in blood (*n* = 10, 11, 14) and sputum (*n* = 8, 9, 13) of control, asthma and COPD subjects. The data are presented as median (column) and individual data. Statistical significance between study groups was labeled as * for *p*-value ≤ 0.05; ** *p*-value ≤ 0.01; *** *p*-value ≤ 0.001.

**Figure 4 cells-12-01631-f004:**
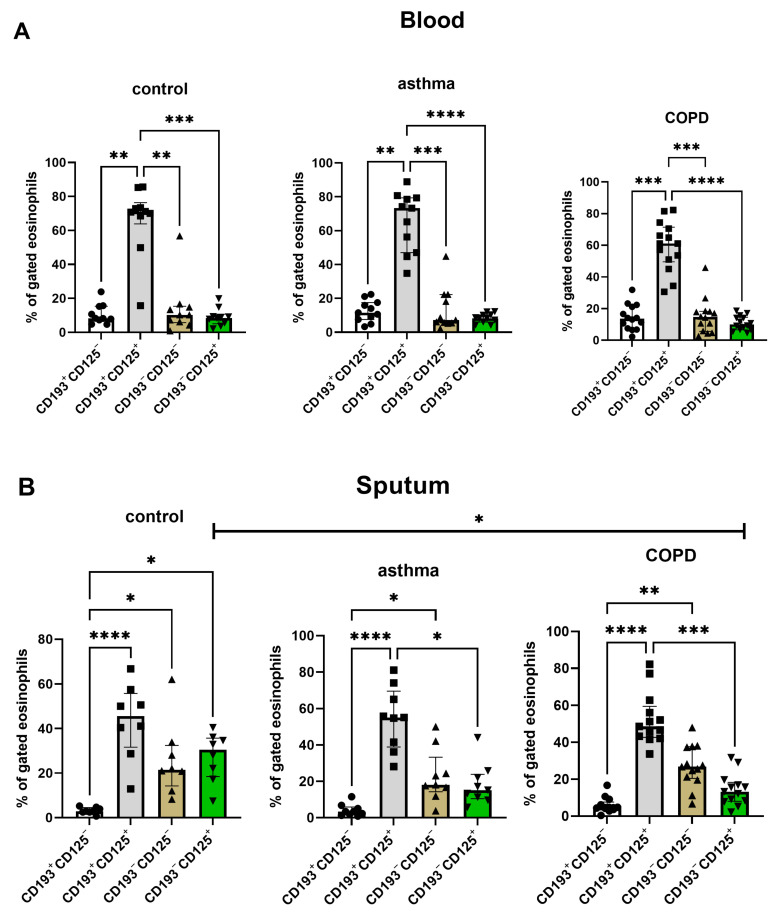
The distribution of blood (*n* = 10, 11, 14) (**A**) and sputum (**B**) (*n* = 8, 9, 13) subpopulations of eosinophils defined by the expression of CD193 and CD125 in control, asthma and COPD groups. The data are presented as median (column), interquartile range (IQR) (whiskers) and individual data (points). Statistical significance between study groups was labeled as * for *p*-value ≤ 0.05; ** *p*-value ≤ 0.01; *** *p*-value ≤ 0.001; **** *p*-value ≤ 0.0001.

**Table 1 cells-12-01631-t001:** Patients’ characteristics.

Variable	Control (*n* = 11)	Asthma (*n* = 14)	COPD (*n* = 15)	*p*-Value
Age (years)	58 (35–63)	46 (39–61)	64 (55–72)	0.046
BMI (kg/m^2^)	28.3 (24.1–32.3)	26 (23–32.5)	27.5 (23–32)	0.832
Females, *n* (%)	6 (55%)	10 (71%)	6 (40%)	0.47
Atopy, *n* (%)	5 (45%)	10 (71%)	4 (27%)	0.049
FEV_1_ (% pred.)	89 (86–102)	89 (84–95)	66 (59–84)	0.002
FEV1/FVC%	71 (70–78)	74 (64–76)	53 (41–62)	<0.001
COPD severity according to GOLD 2020				
GOLD A, *n* (%)	n/a	n/a	4 (27%)	
GOLD B, *n* (%)	n/a	n/a	10 (68%)	
GOLD C, *n* (%)	n/a	n/a	1 (7%)	
Asthma severity according to GINA 2019				
GINA 1, *n* (%)	n/a	9 (64%)	n/a	
GINA 2, *n* (%)	n/a	2 (14%)	n/a	
GINA 3, *n* (%)	n/a	3 (21%)	n/a	
Total IgE (IU/mL)	73.6 (11.3–270.1)	49.2 (33.6–150.1)	30.7 (8.8–79.5)	0.848
CRP (mg/L)	2.9 (2.5–3.2)	2.5 (0.4–4.7)	1.9 (0.85–3.8)	0.834
Blood eosinophil count (cells/µL)	171 (55–185)	219 (137–390)	175 (127–282)	0.554
Blood eosinophil (%)	2 (1–2)	3 (4–7)	2 (1–5)	0.019
Sputum eosinophil (%)	1 (0–1)	1 (0–1)	1 (0–1)	0.88

Data are presented as median (IQR) or number (percentage). Abbreviations: COPD—chronic obstructive pulmonary disease, CRP—C-reactive protein, FEV_1_—forced expiratory volume in one second, IgE—immunoglobulin E.

**Table 2 cells-12-01631-t002:** Sputum cytokine concentrations in control, asthma and COPD subjects.

	Control (*n* = 9)	Asthma (*n* = 12)	COPD (*n* = 14)	*p*-Value
IL-5 (pg/mL)	1.95 (1.61–2.27)	1.84 (1–1.99)	2.16 (0.91–2.91)	0.48
Eotaxin-3 (pg/mL)	0.22 (0–0.6)	0 (0–0.13)	0.26 (0–0.7)	0.17
IL-13 (pg/m)	0 (0–0)	0 (0–0)	0 (0–0)	0.95
CCL5 (pg/mL)	0 (0–0)	0 (0–0)	0 (0–0)	0.09

## Data Availability

The original contributions presented in the study are included in the article. Further inquiries can be directed to the corresponding author.

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
