# Peer review of "Blood and Sputum Eosinophils of COPD Patients Are Differently Polarized than in Asthma"

_cells, 2023, doi:10.3390/cells12121631_

Round 1
Reviewer 1 Report
This study by Mycroft et al., examined the eosinophil phenotypes in COPD and asthma patient’s blood and sputum. By using flow cytometry, authors investigated the different surface markers of eosinophils. Additionally, authors analyzed cytokine, eotaxin levels in sputum by ELISA. The major findings of this study were blood eosinophil markers showed comparable expression among the three groups. However, COPD patients showed higher C193, CD66b sputum eosinophils compared to healthy controls.
The major shortcoming of the manuscript is that it did not add any new information to the field. Authors failed to make a clear statement on the rationale for comparing COPD and asthma. The charestirestic feature of the eosinophils is their high-SSC, due the granularity as authors stated in line 141. However, Fig 1-B (CD16 vs SSC-A) shows opposite. What is the base for eosinophils classification? Is it possible that CD193+ eosinophils can also express CD11b? if so, what is criteria for gating for CD11b vs CD14? And for CD66b vs CD62L. Fig 3-mean value for CD11b for Asthma and COPD sputum shows no difference, however, authors showed significance?
Why authors did not analyze the MFI of the selected markers rather than cell percentage? A simple gating of eosinophils or major sub group then look for expression of markers as presented by MFI would be way simpler and easy for the readers to understand the data.
Overall, I found all graphs are extremely confusing and authors failed to describe them in a read-through style. I suggest authors to analyze and present blood and sputum eosinophils separately (as shown in fig.6).
Minor: the MS writing style is confusing, English editing is highly needed. I found the abstract is incomplete and less interesting. For eg. Authors did not mention about ELISA in method section. Line 21/22 authors stated ‘expression’ of the markers? I did not find any data related to expression.
This is a poorly presented and written MS.
Extensive editing is needed.
Author Response
Dear Editor and Reviewers,
Thank you for your insightful review of our paper “Blood and sputum eosinophils of COPD patients are differently polarized than in asthma” and for giving us the opportunity to submit a revised version of the manuscript to Cells. We very much appreciate the time and effort you put into improving the quality of our manuscript. The revised version of the paper was prepared strictly according to your suggestions. The changes made to the manuscript include the incorporation of new figures, reformulation of abstract, the addition to the content of introduction and some other modifications. All changes were made using the "Track Changes" function in Microsoft Word; the new sections are marked in red. We believe all these refinements resulted in the improvement of the manuscript quality. Thus, we sincerely hope that the manuscript, in its revised form, will be suitable for publication in the journal.
Reviewer 1
This study by Mycroft et al., examined the eosinophil phenotypes in COPD and asthma patient’s blood and sputum. By using flow cytometry, authors investigated the different surface markers of eosinophils. Additionally, authors analyzed cytokine, eotaxin levels in sputum by ELISA. The major findings of this study were blood eosinophil markers showed comparable expression among the three groups. However, COPD patients showed higher C193, CD66b sputum eosinophils compared to healthy controls.
The major shortcoming of the manuscript is that it did not add any new information to the field.
Authors’ response: in the past years, the concept of eosinophil phenotype and function variability has been widely studied in different tissues and in several diseases, including asthma. Eosinophilic and Th2 inflammation is typically present in asthma, whereas in COPD, it is present in only 30-40% of patients and its role in COPD pathogenesis is unclear. The study was conducted to assess blood and sputum eosinophils of COPD patients and to compare it with eosinophils in asthma patients, in whom it has been extensively studied. We hypothesized that eosinophils in COPD and asthma are differently polarized. We are not aware of any study that examined surface marker expression on eosinophils in COPD patients.
Authors failed to make a clear statement on the rationale for comparing COPD and asthma. The charestirestic feature of the eosinophils is their high-SSC, due the granularity as authors stated in line 141. However, Fig 1-B (CD16 vs SSC-A) shows opposite. What is the base for eosinophils classification? Is it possible that CD193+ eosinophils can also express CD11b? if so, what is criteria for gating for CD11b vs CD14? And for CD66b vs CD62L. Fig 3-mean value for CD11b for Asthma and COPD sputum shows no difference, however, authors showed significance?
Authors’ response: We want to thank the Reviewer for this remark. The eosinophils were selected from leukocytes due to their increased granularity. The SSC of selected granulocytes was higher than SSC of the gathered population of monocytes and leukocytes – small cells clearly seen in the third graph in Figure 1b and Figure 1c. The criteria for eosinophil selection were increased SSC and FSC, and lack of CD16 expression. The clearer and corrected Figure 1b and Figure 1c are upgraded in the new version of the manuscript.
Taking into consideration the Reviewer’s critics we withdrew the analysis and comparison of CD62L vs CD66b and CD11b vs CD14 of selected cells. The new version of manuscript includes the improved version of Figure 3 where the comparison of CD125, CD193, CD62L, CD66b, CD14, CD11b expression on eosinophils in blood and sputum of control, asthma, and COPD subjects. This figure presents the differences in marker expression between blood and sputum eosinophils and presents the discrepancy of eosinophil polarization for CD193, CD66b and CD11b expression in eosinophils from COPD patients. Although is not spectacularly seen in Figure 3 but the significantly decreased level of CD11b+ sputum eosinophils in COPD compared to asthma (97.9 [96.6-99.1] % vs 99.4 [98.9-100] %, respectively, p=0.009) is correct and confirmed by additional statistical analysis.
We also did not change the Figure 4 - The distribution of blood and sputum subpopulations of eosinophils defined by the expression of CD193 and CD125 in control, asthma, and COPD groups.
Why authors did not analyze the MFI of the selected markers rather than cell percentage? A simple gating of eosinophils or major sub group then look for expression of markers as presented by MFI would be way simpler and easy for the readers to understand the data.
Authors’ response: We did not analyse the marker expression using MFI due to significant discrepancy in numbers of cells gated in the final eosinophil gate. During acquisition at least 50 000 cells in the P1 - singlet gate were collected. However, in the sputum samples much less cells were collected, this type of material is characterized by a high number of debris and lower eosinophil number than blood samples. Moreover, it was very difficult to find controls with increased blood eosinophilia, in this group sputum eosinophilia was extremely difficult to catch. In face of different counts of eosinophils collected from blood and sputum samples as well as in patients and controls samples we did not analyse the data according to MFI due to possible errors in results interpretation choosing % of gated eosinophils as a more optimized method.
Overall, I found all graphs are extremely confusing and authors failed to describe them in a read-through style. I suggest authors to analyze and present blood and sputum eosinophils separately (as shown in fig.6).
Authors’ response: We corrected Figure 3 as suggested by the Reviewer. We hope this type of data presentation is easier for the reader.
Minor: the MS writing style is confusing, English editing is highly needed. I found the abstract is incomplete and less interesting. For eg. Authors did not mention about ELISA in method section. Line 21/22 authors stated ‘expression’ of the markers? I did not find any data related to expression.
Authors’ response: we performed English language editing. We have modified the abstract, we added the information about ELISA to the methods and we added some data to the results section.
The English style of manuscript was corrected, the abstract was improved, and additional information was added.
This is a poorly presented and written MS.

Reviewer 2 Report
The manuscript “Blood and sputum eosinophils of COPD patients are differently polarized than in asthma.” The manuscript aimed to evaluate the profile of eosinophil subpopulations in patients with COPD, since the profile of eosinophils could imply changes in the clinical treatment of patients.
The work would benefit overall if the results were presented more clearly and objectively.
- Figure 1 presents the values of the results are very small, which makes it difficult to read and interpret the gating strategy performed, it is important to change the layout of the figure.
- The caption of Figure 1 presents little detail of the information presented. A legend with more details is required.
- The caption of figure 2 needs to be better detailed, what are the criteria for excluding materials, blood and sputum?
Author Response
Dear Editor and Reviewers,
Thank you for your insightful review of our paper “Blood and sputum eosinophils of COPD patients are differently polarized than in asthma” and for giving us the opportunity to submit a revised version of the manuscript to Cells. We very much appreciate the time and effort you put into improving the quality of our manuscript. The revised version of the paper was prepared strictly according to your suggestions. The changes made to the manuscript include the incorporation of new figures, reformulation of abstract, the addition to the content of introduction and some other modifications. All changes were made using the "Track Changes" function in Microsoft Word; the new sections are marked in red. We believe all these refinements resulted in the improvement of the manuscript quality. Thus, we sincerely hope that the manuscript, in its revised form, will be suitable for publication in the journal.
Reviewer 2
The manuscript “Blood and sputum eosinophils of COPD patients are differently polarized than in asthma.” The manuscript aimed to evaluate the profile of eosinophil subpopulations in patients with COPD, since the profile of eosinophils could imply changes in the clinical treatment of patients.
The work would benefit overall if the results were presented more clearly and objectively.
- Figure 1 presents the values of the results are very small, which makes it difficult to read and interpret the gating strategy performed, it is important to change the layout of the figure.
Authors’ response: As suggested, Figure 1 was changed and improved – we withdrew the analysis of eosinophils with evaluated markers, we hope this figure is clearer for the reader.
- The caption of Figure 1 presents little detail of the information presented. A legend with more details is required.
Authors’ response: Figure 1. The gating strategy. (a) fluorescent-minus-one (FMO) (b) blood, (c) sputum.
Legend: Fluorescence-minus-one (FMO) controls for the experiment. FMO controls were prepared without adding a particular fluorophore-conjugated isotype control antibody as shown in table below.
Gating strategy: Obtained events were gated in an FSC-A vs FSC-H intensity dot plot to eliminate doublets. Leukocytes were gated on CD45+ vs FSC. Granulocytes were selected as cells with high FSC and high SSC. The eosinophils were selected from granulocytes as CD16- cells. Forward scatter, SSC: Side scatter.
- The caption of figure 2 needs to be better detailed, what are the criteria for excluding materials, blood and sputum?
Authors’ response: we have added more details to the caption of Figure 2. 40 patients were recruited, All patients had undergone blood sampling and sputum induction. Overall, 5 blood samples were not taken into FACS analysis because of low eosinophil levels and 10 sputum samples because of poor sputum quality (>20% of squamous cells, low number of cells) or low eosinophil levels. Blood samples were excluded if there were no eosinophils present in flow cytometry. The sputum was excluded if :1) the percentage of squamous cells was >20%, 2) there was a low number of cells isolated from sputum <1x106, 3) there were no eosinophils in flow cytometry.

Round 2
Reviewer 1 Report
I appreciate authors' efforts in improving the MS. I have no further comments.